# High Probability Contextual Bandits for Optimal Dosage Selection

## Abstract

Multi-Armed Bandit (*MAB*) formulations are commonly used to model the problem of *Optimal Dose-Finding*. However, in many practical applications, it is necessary to receive data about the patient's current state and then administer a drug dosage adapted to that state. To overcome this issue, we adopt a linear contextual bandit formulation with stage-wise constraints. At each round, the learner selects a dosage and receives both a reward signal and a cost signal. The learner's goal is to maximize the drug's efficacy—captured as the expected cumulative reward—while ensuring that the toxicity, reflected by the cost signal, remains below a known threshold. Satisfying the cost signal constraint only in expectation can be dangerous, as it may lead to over-dosage complications in certain cases. To address this issue, we introduce a novel model that controls the realization of the cost signal with high probability, in contrast to previous works where control was only applied to the expected cost signal. Our algorithm follows the *UCB* approach, for which we establish a regret bound over $T$ rounds and run numerical experiments. We further generalize our results to *non-linear* functions and provide a regret bound in terms of the *eluder dimension*, a measure of function class complexity.

## 1 Introduction

Ensuring patient safety while maximizing therapeutic efficacy is a critical challenge in medical treatments, especially in optimal drug dosing where overdosing can have serious consequences. In these scenarios, the objective is to find an appropriate treatment regime, while ensuring patients are not exposed to dangerous drug dosages. Online learning methods, widely employed in various decision-making systems, offer promising solutions to this challenge by adapting decisions based on observed outcomes (Matsuura et al. (2022),Lee et al. (2020)).

This framework is extensively applied in Optimal Dose-Finding Huckvale et al. (2023); Shen et al. (2020); Chen & Khademi (2024), where adaptive methods aim to determine the optimal drug dosage that maximizes therapeutic efficacy while safeguarding patients from overdosing. However, most existing approaches are setting a binary variable specifying the treatment's outcome , which fails to capture the nuances of these signals. Moreover, in order to guarantee drug dosages do not cause dangerous side effects, current studies have proposed algorithms that control the expected toxicity per time step (Pacchiano et al. (2024),Amani et al. (2019),Amani et al. (2021)). Unfortunately, satisfying this solution concept may not prevent the harmfulness of the drug dosages administered to many individual patients. Consequently, there remains a risk of overdosing, posing serious health risks to patients. Additionally, not all approaches incorporate contextual information, despite its importance in medical treatments where dosage decisions are often based on patient-specific tests or conditions.

In the bandit literature, constraints are typically based either on the history of previous rewards (e.g., knapsack bandits Badanidiyuru et al. (2018), fairness constraints Joseph et al. (2016)) or involve simultaneous reward and cost signals, with constraints depending on the cost, as in Amani et al. (2019); Pacchiano et al. (2021; 2024). The latter formulation is particularly useful for modeling applications advertising, and drug dosing. Problems in this domain can be cast in the reward/cost model of Pacchiano et al. (2021), the drug dosage administered generates a reward signal (efficacy) and a cost signal (toxicity). This is precisely the scenario encountered in Phase II clinical trials,

where physicians aim to adaptive adjust dosages to maximize efficacy while ensuring toxicity remains below a safe threshold $\tau$.

Previous works on contextual bandits with stage-wise constraints Pacchiano et al. (2024) propose contextual bandits algorithms that ensure the expected cost remains below a safe threshold $\tau$. However, this does not prevent individual instances where the cost exceeds the threshold, which can be unacceptable in critical applications like medicine. Therefore, there is a pressing need for models that guarantee, with high probability, that the realized cost stays within safe limits in each instance.

In this paper, we address this need by developing a *UCB-like* algorithm, which we call *High Probability Constrained UCB*. Our algorithm is grounded in the *Optimism in the Face of Uncertainty* (OFU) principle Auer (2002) and does not require prior knowledge of any initial safe dosage—a significant advantage over existing methods. It operates effectively under both adversarial and stochastic contexts and relies only on the standard assumption of sub-Gaussian noise distributions. By leveraging this assumption, we formulate a constraint event that ensures, with high probability, that the cost signal remains below the desired threshold in each round.

Furthermore, we note that knowing the exact noise distribution allows for even tighter constraints, ensuring the cost realization stays within safe limits with high probability. Our approach accounts for the randomness in constraint satisfaction due to noise and is validated with high probability based on past observations of contexts, rewards, and costs. Another approach to this problem is *Distributional Reinforcement Learning*, where noise distributions are characterized, and instance-dependent bounds are derived Wang et al. (2023).

Through rigorous analysis, we prove that our algorithm achieves a $T$-round regret bound of order $\tilde{\mathcal{O}}(d\sqrt{T})$. We also demonstrate the practical effectiveness of our approach through numerical experiments. Additionally, we extend our results to settings with *non-linear* reward and cost functions by parameterizing our analysis in terms of the *eluder dimension*, a complexity measure for function classes.

**Contributions** In this work, we introduce a high-probability model for dose finding in the contextual bandit framework, which, to the best of our knowledge, has not been explored in the literature. Despite the crucial importance of controlling realized adverse effects during clinical interventions, existing models and algorithms primarily focus on controlling expected effects. One of the main contributions of this work is the introduction of a more realistic high-probability model that addresses this gap. Additionally, we have developed algorithms with provable guarantees for this setting. Our contributions are both conceptual and technical, offering a new perspective on dose-finding methodologies within contextual bandits.

## 2 PROBLEM FORMULATION

**Notation.** We adopt the following notation throughout the paper. We denote by $\langle x, y \rangle = x^\top y$ and $\langle x, y \rangle_{\mathbf{A}} = x^\top \mathbf{A} y$, for a positive definite matrix $\mathbf{A} \in \mathbb{R}^{d \times d}$, the inner-product and weighted inner-product of vectors $x, y \in \mathbb{R}^d$. Similarly, we denote by $\|x\| = \sqrt{x^\top x}$ and $\|x\|_{\mathbf{A}} = \sqrt{x^\top \mathbf{A} x}$, the $\ell_2$ and weighted $\ell_2$ norms of vector $x \in \mathbb{R}^d$. We use upper-case letters for random variables (e.g., $X$), and their corresponding lower-case letters for a particular instantiation of that random variable (e.g., $X = x$). The set $\{1, \ldots, T\}$ is denoted by $[T]$. Finally, we use $\tilde{\mathcal{O}}$ for the big-$\mathcal{O}$ notation up to logarithmic factors.

Our goal is to design an algorithm for the adaptive dosage allocation problem. The goal of the algorithm is to maximize the expected efficacy of the drug while preserving low toxicity levels with high probability. We choose to study the following *constrained contextual linear bandit* to approach this problem. In each round $t \in [T]$ the agent is observing a $d$-dimensional vector $X_t \in \mathbb{R}^d$ that represents medical tests or results about the patient that are applied each day before the drug intake. The nature of the context can be adversarial or stochastic, our algorithm works for both. Then, the agent selects a dosage $\alpha_t \in [0, 1]$. We assume that a dosage cannot take an infinite value so we can normalize it in $[0, 1]$. After the dosage selection, the learner observes a pair $(R_t, C_t)$ that denotes a reward and a cost signal. Their mathematical expressions are $R_t = \alpha_t \langle X_t, \theta^* \rangle + \alpha_t \xi_t^r$ and $C_t = \alpha_t \langle X_t, \mu^* \rangle + \alpha_t \xi_t^c$. The unknown vectors $\theta^*$ and $\mu^*$ may represent the importance of each of the selected features to the efficacy and the toxicity that the drug provokes to them. Moreover,

the inner product is a measure of the similarity of the symptoms $X_t$ and the sickness. For example, if $\langle X_t, \theta^* \rangle > 0$ then the patient suffers from the illness and needs a positive dosage from the drug. Otherwise, there is no reason for the patient to take the drug. Without the presence of noise, we assume that the effects of both the efficacy and the toxicity are an increasing function of the dosage $\alpha_t$. This assumption is commonly adopted in the dosage allocation bibliography as cited in Chevret (2006).

The protocol can be summarized as follows. At each round $t$: (1) observe *context* $X_t \in \mathbb{R}^d$, (2) select *dosage* $\alpha_t \in [0, 1]$, (3) observe the reward $R_t \in \mathbb{R}$ and the constraint $C_t \in \mathbb{R}$.

Before proceeding with our analysis and results, we first outline the standard assumptions for the model. These assumptions are well-established in the literature on contextual bandits with constraints.

**Assumption 1** (Sub-Gaussian noise). *For all $t \in [T]$, the reward and cost noise random variables $\xi_t^r$ and $\xi_t^c$ are conditionally Sub-Gaussian, i.e., for all $\alpha \in \mathbb{R}$,*

$$\mathbb{E}[\xi_t^r \mid \mathcal{H}_{t-1}] = 0, \quad \mathbb{E}[\exp(\alpha\xi_t^r) \mid \mathcal{H}_{t-1}] \leq \exp(\alpha^2\gamma_r^2/2),$$

$$\mathbb{E}[\xi_t^c \mid \mathcal{H}_{t-1}] = 0, \quad \mathbb{E}[\exp(\alpha\xi_t^c) \mid \mathcal{H}_{t-1}] \leq \exp(\alpha^2\gamma_c^2/2),$$

*where $\mathcal{H}_t$ is the filtration that includes all the events $(R_{1:t}, C_{1:t}, \xi_{1:t}^r, \xi_{1:t}^c)$ until the end of round $t$.*

**Assumption 2** (bounded parameters). *There is a known constant $S > 0$, such that $\|\theta_*\| \leq S$ and $\|\mu_*\| \leq S$.[1]*

**Assumption 3** (bounded actions). *The $\ell_2$-norm of all contexts are bounded by $L > 0$, i.e.,*

$$\max_{t \in [T]} \|X_t\| \leq L.$$

**Assumption 4** (Positive toxicity threshold). *The toxicity constraint in order to be meaningful must satisfy that $\tau > 0$.*

We observe that our analysis does not require knowledge of an initial safe dosage, unlike in (Pacchiano et al., 2024) or any assumption about the initial decision set like in (Moradipari et al., 2019). However, we believe that in their analysis, this assumption can be relaxed because the vector $\mu^*$ is bounded, and any $X_t$ from their decision set satisfying $\|X_t\| \leq \frac{\tau}{S}$ can serve as an initial safe dosage. They mention this possibility in their related works. This follows from the inequality $\langle X_t, \mu^* \rangle \leq \|X_t\| \|\mu^*\| \leq \frac{\tau}{S} S = \tau$.

In each round $t$, the agent is constrained to select a dosage $\alpha_t$ such that $\alpha_t \left( \langle X_t, \mu^* \rangle + \gamma_c \sqrt{2 \log\left(\frac{1}{\delta}\right)} \right) \leq \tau$. We demonstrate in Section 4 that when this constraint is satisfied, it ensures $\mathbb{P}_{\xi_t^c}(C_t \leq \tau) \geq 1 - \delta$. We define the set of feasible dosages as

$$\mathcal{A}_t^f = \left\{ \alpha \in [0, 1] : \alpha \left( \langle X_t, \mu^* \rangle + \gamma_c \sqrt{2 \log\left(\frac{1}{\delta}\right)} \right) \leq \tau \right\}.$$

Due to the unknown nature of $\mu^*$, this set is initially uncertain, and we must estimate it.

Maximizing the expected reward over $T$ rounds is equivalent to minimizing the *expected $T$-round constrained pseudo-regret*, defined as

$$\mathcal{R}_\mathcal{C}(T) = \sum_{t=1}^{T} (\alpha_t^* - \alpha_t)\langle X_t, \theta^* \rangle, \tag{1}$$

where $\alpha_t^*$ is the *optimal feasible dosage* in round $t$, i.e., $\alpha_t^* \in \arg\max_{\alpha \in \mathcal{A}_t^f} \alpha\langle X_t, \theta^* \rangle$, and $\alpha_t$ is the dosage selected by the agent in round $t$, which belongs to the set of feasible actions in that round, i.e., $\alpha_t \in \mathcal{A}_t^f$, with high probability over $\xi_{1:t-1}^c, C_{1:t-1}$.

## 3 RELATED WORK

Multi-armed bandits for adaptive optimal dosage finding have been widely used in the literature over the past two years. In Shen et al. (2020), they consider a discrete set of $K$ doses. For each dose

---

[1]The choice of the same upper-bound $S$ for both $\theta_*$ and $\mu_*$ is just for simplicity and convenience.

$d_k$, where $k \in [K]$, there exist two unknown parameters $q_k$ and $p_k$. These parameters characterize the probabilities of the drug being effective and toxic, respectively. More precisely, let $X$ and $Y$ be Bernoulli random variables with parameters $q_k$ and $p_k$, respectively.

To correlate the dose and toxicity, they adopt a model from O'Quigley et al. (1990), where the relation is given by:

$$p_k(\alpha) = \left( \frac{\tanh(d_k) + 1}{2} \right)^\alpha,$$

where $\alpha$ is a global parameter across all dose levels that needs to be estimated.

In our opinion, there are several issues with this model. First, the reward and cost signals take discrete values, failing to capture intermediate responses that better model real-world scenarios. Moreover, their model for the cost signal is too specific and may not capture the toxicity profiles of a wider variety of drugs. Finally, their guarantee concerns the empirical mean of the cost being below the desired threshold with high probability. However, this guarantee does not exclude the possibility that at least one patient may be overdosed, potentially leading to serious health consequences.

In (Huckvale et al., 2023), they adopt a contextual perspective to dosage allocation problems. Various models in the literature (Chen & Khademi (2024), Chevret (2006)) have followed similar assumptions to ours, namely that the increase of the dosage is proportional to the increase of the expected value of the reward and cost. Moreover, since our model is stochastic, we assume that increasing the dosage proportionally affects the noise of the signal. Analyzing in more detail, higher dosages may not contribute to the drug's efficacy as much as expected, while lower dosages tend to be less effective but produce more consistent results.

In linear bandits under stage-wise constraints, multiple works (Pacchiano et al. (2024), Amani et al. (2021), Moradipari et al. (2019)) have formulated constraints that can control the expected value of the cost signal. One possible approach to extend these works to control the realization of the signal with high probability is to lower the value of the threshold. However, it is not yet clear by how much it should be lowered and whether this strategy could potentially control the behavior of the tails of the noise distribution. Moreover, in Pacchiano et al. (2024), their algorithm requires an initial safe action, whereas ours can take advantage of standard assumptions to prove that there always exists an initial safe dosage.

Finally, in the work of (Wang et al., 2023), which belongs to the field of Distributional Reinforcement Learning, they provide *instance-dependent* bounds while assuming that the distribution of the noise belongs to a known class.

## 4 CONSTRAINT FORMULATION

The learner's goal is to maximize the cumulative reward or efficacy of the drug administered to the patient over $T$ rounds, i.e., $\sum_{t=1}^{T} \alpha_t \langle X_t, \theta^* \rangle$, while ensuring that the realized toxicity remains below a known threshold with high probability. In clinical trials terminology, this problem is referred to as a Phase 2 trial, as the toxicity threshold $\tau$ is considered known in advance. Our algorithm takes as input a confidence level $\delta$ to control the realization of the noise. To model the requirement of controlling the cost realization with high probability, we impose a nonlinear constraint involving $\delta$. It should be noted that if we know the exact distribution of the noise, this problem can be solved exactly without introducing this constraint by using a similar algorithm.

We formulate the following constraint:

**Lemma 4.1.** *When the selected dosage $\alpha_t$ satisfies $\alpha_t \left( \langle X_t, \mu^* \rangle + \gamma_c \sqrt{2 \log \left( \frac{1}{\delta} \right)} \right) \leq \tau$ then it holds that $\Pr(C_t \leq \tau \mid \mathcal{H}_t) \geq 1 - \delta$.*

*Proof.* Provided in the A.1. $\qquad \square$

We note that, given the distribution of the noise at round $t$, $\xi_t^c$, it holds that $\Pr(C_t \leq \tau \mid \mathcal{H}_t) \geq 1 - \delta$. The constraint

$$\alpha_t \left( \langle X_t, \mu^* \rangle + \gamma_c \sqrt{2 \log \left( \frac{1}{\delta} \right)} \right) \leq \tau$$

is thus satisfied with high probability with respect to $\mathcal{H}_{t-1}$.

An advantage of our model compared to previous ones (Pacchiano et al., 2024) is that, under the standard assumptions of linear contextual bandits, it does not require knowledge of an *initial safe dosage*. Initially, the learner knows nothing about the unknown vectors. To gain information from the reward and cost signals, the learner needs to select a non-zero dose. We demonstrate that our initial assumptions imply the existence of an initial safe dose.

Since $\tau > 0$, we show that an initial safe interval for choosing $\alpha_1$ is

$$\left[ 0, \min\left( 1, \frac{\tau}{\gamma_c \sqrt{2 \log\left(\frac{1}{\delta}\right)} + LS} \right) \right].$$

To begin with, if

$$\langle X_t, \mu^* \rangle + \gamma_c \sqrt{2 \log\left(\frac{1}{\delta}\right)} \leq 0,$$

then $\mathcal{A}_0^f = [0, 1]$. Otherwise, $\alpha_0$ can range from $0$ up to

$$\min\left( 1, \frac{\tau}{\gamma_c \sqrt{2 \log\left(\frac{1}{\delta}\right)} + LS} \right),$$

since $\langle X_t, \mu^* \rangle \leq LS$ by the Cauchy–Schwarz inequality.

This is an important factor in the design of dosage allocation, as it eliminates the risk of overdosing the patient. Moreover, it reduces the number of hyper-parameters that need to be tuned before running the algorithm.

### 4.1 ALGORITHM

We aim for our algorithm to adhere to the fundamental principle of *Optimism in the Face of Uncertainty* (OFU). Additionally, we need to make robust choices to ensure that the constraint is satisfied with high probability. To achieve this, we intend to be optimistic in our estimates for the reward signal and pessimistic for the cost.

In each non-zero dose round, we construct two least squares estimators: one for $\theta^*$ and one for $\mu^*$. For a given regularization parameter $\lambda > 0$, the regularized covariance matrix at round $t$ is defined as:

$$\Sigma_t = \lambda I + \sum_{s=1}^{t-1} X_s X_s^\top. \tag{2}$$

Using equation equation 2, we define the regularized least squares estimators $\hat{\theta}_t$ and $\hat{\mu}_t$.

$$\hat{\theta}_t = \Sigma_t^{-1} \sum_{s:\alpha_s \neq 0} \alpha_s^{-1} R_s X_s \quad \hat{\mu}_t = \Sigma_t^{-1} \sum_{s:\alpha_s \neq 0} \alpha_s^{-1} C_s X_s \tag{3}$$

We note that we use only the contexts $X_t$ and the corresponding realizations of the reward and cost signals $R_t$ and $C_t$ for the rounds in which we assigned a non-zero dosage. In rounds where we assigned zero dosage, we did not receive feedback about the dosage effect; that is, $R_t = C_t = 0$, given the way our model is constructed.

To design a *UCB-like* algorithm, we need to define high-probability confidence sets centered at our estimators $\hat{\theta}_t$ and $\hat{\mu}_t$. These confidence sets will enable us to derive upper bounds on the distances between our estimators and the unknown vectors $\theta^*$ and $\mu^*$. To construct the desired confidence intervals, we will employ the following fundamental theorem.

**Theorem 4.1** (Thm. 2 in Abbasi-Yadkori et al., 2011). *For a fixed $\delta \in (0, 1)$ and*

$$\beta_t^r(\delta, d) = \gamma_r \sqrt{d \log\left( \frac{1 + (t-1)L^2/\lambda}{\delta} \right)} + \sqrt{\lambda} S, \qquad \forall t \in [T],$$

$$\beta_t^c(\delta, d) = \gamma_c \sqrt{d \log\left(\frac{1 + (t-1)L^2/\lambda}{\delta}\right)} + \sqrt{\lambda}S, \qquad \forall t \in [T],$$

*it holds with probability at least $1 - \delta$ that*

$$\|\widehat{\theta}_t - \theta_*\|_{\Sigma_t} \le \beta_t^r(\delta, d), \qquad \|\widehat{\mu}_t - \mu_*\|_{\Sigma_t} \le \beta_t^c(\delta, d).$$

Using Theorem 4.1, we now define the following confidence sets (ellipsoids):

$$\begin{aligned}
\mathcal{C}_t^r &= \{\theta \in \mathbb{R}^d : \|\theta - \widehat{\theta}_t\|_{\Sigma_t} \le \beta_t(\delta, d)\}, \\
\mathcal{C}_t^c &= \{\mu \in \mathbb{R}^d : \|\mu - \widehat{\mu}_t\|_{\Sigma_t} \le \beta_t(\delta, d)\},
\end{aligned} \tag{4}$$

Theorem 4.1 suggests that $\theta^* \in \mathcal{C}_t^r$ and $\mu^* \in \mathcal{C}_t^c(\alpha_c)$, each with probability at least $1 - \delta$. We will use these confidence intervals to create our estimators for $\theta^*$ and $\mu^*$.

We aim to be optimistic in our estimate of $\theta^*$ by selecting

$$\tilde{\theta}_t = \arg\max_{\theta \in \mathcal{C}_t^r} \langle X_t, \theta \rangle, \tag{5}$$

and pessimistic about $\mu^*$ by choosing $\tilde{\mu}_t$ that minimizes the volume of the estimated feasible set. In our case, the feasible set is a continuous sub-interval of $[0, 1]$, so its measure is simply its length.

Before describing the algorithm, we will first define the confidence ellipsoids and the *Least Squares Estimators* for $\theta^*$ and $\mu^*$.

---

**Algorithm 1** High Probability Constrained UCB

---

1: **Input:** Constraint threshold $\tau \ge 0$; Confidence parameter $\delta$; Sub-Gaussianity constant $\gamma_c$
2: $\alpha_0 \leftarrow \min\{1, \frac{\tau}{\gamma_c\sqrt{2\log\left(\frac{1}{\delta}\right)} + L_X \cdot L_{\mu^*}}\}$
3: **for** $t = 1, 2, \cdots, T$ **do**
4:     Compute $\hat{\mu}$ according to 3
5:     Use $\hat{\mu}$ to compute the estimated feasible set $\hat{\mathcal{A}}_t^f$ using 7
6:     Compute $\hat{\theta}_t$ using equation 5
7:     Compute action $\alpha_t = \arg\max_{\alpha \in \hat{\mathcal{A}}_t^f} \alpha\langle X_t, \hat{\theta}_t \rangle$
8:     Take action $\alpha_t$ and if $\alpha_t \ne 0$ store the reward and the cost signals $(R_t, C_t)$
9: **end for**

---

The computation of the estimated feasible set $\hat{\mathcal{A}}_t^f$ is performed in two steps. First, we estimate the unknown cost vector $\mu^*$ using a least squares estimator. This procedure yields a confidence ellipsoid that contains $\mu^*$ with high probability. Among all $\mu$ within this ellipsoid, we select the one that minimizes the length of the interval of feasible values for $\alpha_t$.

### 4.2 CHOICE OF $\hat{\mu}$

As previously discussed, we aim to choose our estimate—denoted by $\tilde{\mu}$—pessimistically to minimize the length of $\hat{\mathcal{A}}_t^f$. By definition, $\hat{\mathcal{A}}_t^f$ is given by

$$\hat{\mathcal{A}}_t^f = \left\{ \alpha \in [0, 1] : \alpha\left(\langle X_t, \tilde{\mu} \rangle + \gamma_c\sqrt{2\log\left(\frac{1}{\delta}\right)}\right) \le \tau \right\}.$$

Since $\tau > 0$, we first need to check the sign of $\langle X_t, \tilde{\mu} \rangle + \gamma_c\sqrt{2\log\left(\frac{1}{\delta}\right)}$. If this expression is negative for all $\mu \in \mathcal{C}_\mu^t$, then we set $\hat{\mathcal{A}}_t^f = [0, 1]$. However, if there exists a $\mu \in \mathcal{C}_\mu^t$ such that this expression is positive, we select the $\mu$ that minimizes the maximum feasible $\alpha_t$. This approach can be summarized in the following convex program, where $\hat{\mu}$ is the least squares estimate of $\mu$.

$$\begin{aligned}
\max_{\boldsymbol{\mu}} &\quad \langle X_t, \mu \rangle \\
\text{subject to} &\quad \|\mu - \hat{\mu}\|_{V_t} \le b_t^2, \\
&\quad \langle X_t, \mu \rangle + \gamma_c\sqrt{2\log\left(\frac{1}{\delta}\right)} \ge 0
\end{aligned} \tag{6}$$

Let $\mathcal{K}(x)$ the set that contains the solutions of the convex program 6.

$$
\hat{\mathcal{A}}_t^f = \begin{cases} [0,1] & \text{, if } \mathcal{K}(\hat{\mu}_t) = \emptyset \\ [0, \frac{\tau}{\mathcal{K}(\hat{\mu}_t) + \gamma_c \sqrt{2\log\left(\frac{1}{\delta}\right)}}] & \text{, if } \mathcal{K}(\hat{\mu}_t) \neq \emptyset \end{cases} \tag{7}
$$

Before beginning the analysis of our algorithm, we observe that its computational cost is primarily due to solving one convex program for $\tilde{\mu}$ and another for $\tilde{\theta}$. These convex programs have linear objectives; one involves a combination of quadratic and linear constraints, while the other involves only quadratic constraints. We can employ standard optimization algorithms, such as interior-point methods, to solve these sub-problems. In our numerical experiments, we found that they are solved in a short amount of time.

## 5 REGRET ANALYSIS

The objective of the agent is to minimize the *expected $T$-round (constrained) (pseudo)-regret*, i.e.,

$$
\mathcal{R}_{\mathcal{C}}(T) = \sum_{t=1}^{T} r^*(X_t) - r(X_t)
$$

where

$$
r^*(X_t) = \max_{\alpha \in \mathcal{A}_t^f} \alpha \langle X_t, \theta^* \rangle
$$

$$
r(X_t) = \max_{\alpha \in \hat{\mathcal{A}}_t^f} \alpha \langle X_t, \theta^* \rangle
$$

We see that the choice of $\alpha$ depends on the sign of $\langle X_t, \theta^* \rangle$. If this inner product is positive we choose the largest feasible value and otherwise the lowest feasible one.

$$
\begin{aligned}
\mathcal{R}_{\mathcal{C}}(T) &= \sum_{t=1}^{T} \max_{\alpha \in \mathcal{A}_t^f} \left(\alpha \langle X_t, \theta^* \rangle\right) - \alpha_t \langle X_t, \theta^* \rangle \\
&= \sum_{t=1}^{T} (\alpha_t^* - \alpha_t) \langle X_t, \theta^* \rangle
\end{aligned} \tag{8}
$$

We will use a decomposition of the regret similar to standard ones in the *Linear Bandits under constraints* literature, (Amani et al. (2019),Pacchiano et al. (2021),Pacchiano et al. (2024)). We define as

$$
\tilde{\alpha}_t = \underset{\alpha \in \mathcal{A}_t^f}{\arg\max} \{\alpha \langle X_t, \hat{\theta}_t \rangle\} \tag{9}
$$

Using the above definition we decompose the regret as follows.

$$
\begin{aligned}
\mathcal{R}_{\mathcal{C}}(T) &= \sum_{t=1}^{T} (\alpha_t^* - \alpha_t) \langle X_t, \theta^* \rangle \\
&= \underbrace{\sum_{t=1}^{T} (\alpha_t^* - \tilde{\alpha}_t) \langle X_t, \theta^* \rangle}_{\text{Term 1: Cost for approximating } \theta^*} + \underbrace{\sum_{t=1}^{T} (\tilde{\alpha}_t - \alpha_t) \langle X_t, \theta^* \rangle}_{\text{Term 2: Cost for approximating } \mu^*}
\end{aligned} \tag{10}
$$

### 5.1 ANALYSIS OF THE REGRET

**Lemma 5.1.** *The first term in the regret decomposition can be bounded as follows:*

$$
(\alpha_t^* - \tilde{\alpha}_t) \langle X_t, \theta^* \rangle \leq \tilde{\alpha}_t \langle X_t, \tilde{\theta} - \theta^* \rangle
$$

*Proof.* The proof is in A.2. □

We note that this is the standard bound in the Linear Bandits literature as first proved in the classical work of (Abbasi-Yadkori et al., 2011). It remains to bound the second term.

**Lemma 5.2.** *The first term in the regret decomposition can be bounded as follows:*

$$(\tilde{\alpha}_t - \alpha_t)\langle X_t, \theta^* \rangle \leq L \cdot S \cdot \frac{\langle X_t, \tilde{\mu} - \mu^* \rangle}{\tau}$$

*Proof.* The proof is in A.3. $\square$

By using 5.1, 5.2 combining with the regret decomposition 10 we can bound the regret as following. The bound is conditioned on the following event that holds with probability at least $1 - \delta'$.

$$\mathcal{E} := \left\{ \|\tilde{\theta}_t - \widehat{\theta}_t\|_{\Sigma_t} \leq \beta_t(\delta', d) \wedge \|\tilde{\mu}_t - \widehat{\mu}_t\|_{\Sigma_t} \leq \beta_t(\delta', d) \right\} \tag{11}$$

It is important to mention that $\delta'$ is not necessary equal to $\delta$. The first one is the probability that the regret bounds holds and the second one the probability that the realization of the noise of the cost stays below the threshold.

**Theorem 5.1.** *With probability at least one $1 - \delta'$ the regret 10 can be bounded by* $\mathcal{O}\left( \beta_T(\delta', d) \sqrt{2Td \log\left(1 + \frac{TL^2}{\lambda}\right)} \right)$.

*Proof.* The proof is in A.4. $\square$

## 6 EXPERIMENTAL RESULTS

Ideally, we had hoped to test our algorithms on real data, such as patient data or cell cultures. However, due to limited access to real data, we decided to evaluate our algorithms using synthetic data. Before analyzing our procedure, we should note that in the linear model, the dot product can take both positive and negative values. Therefore, we expect that the expected regret plot, in addition to exhibiting sub-linear behavior, may also show decreasing behavior.

To generate $\theta^*$ and $\mu^*$, we sampled them from a $d$-dimensional normal distribution and then normalized them. The contexts were also generated from a multivariate normal distribution and normalized accordingly. We ran our experiments with 5 and 10-dimensional vectors, using multiple values of $\tau$ and for $10^5$ iterations. In practice, it is interesting to examine the relationship between $\tau$ and $\max \|X\|$, $\|\theta^*\|$, and $\|\mu^*\|$.

We observed that for small values of $\tau$, such as $0.2$, the set of feasible dosages has a very small length, and the results are dominated by noise. However, for larger values of $\tau$, like $0.6$ and $0.8$, the results showed that the sub-linear behavior was already visible after $10^4$ iterations. Remarkably, after $4 \times 10^5$ iterations, we noticed decreasing behavior, which is a sign that our estimators have converged to the true $\theta^*$ and $\mu^*$.

We also plotted the number of times the constraint was violated when we satisfied the constraint in expectation. We plotted this over $10^4$ rounds, and the behavior was linear, indicating that until our estimators are close to the true vectors, we overdose the patients multiple times. We ran our experiments 10 times and plotted the mean value and the standard deviation around the regret.

We hope that in the future we could extend our experiments in real data like cell cultures so we can check good choices for the vectors $\theta^*, \mu^*$, good features-contexts, an appropriate value for $\tau$ and non-linear models too.

## 7 NON-LINEAR REWARDS AND COSTS

Instead of modeling the reward and the cost signal as linear functions in term of the unknown parameters $\theta$ and $\mu$ we can use more general functions and express our results in terms of the *Eluder dimension* as defined in Russo & Van Roy (2013).

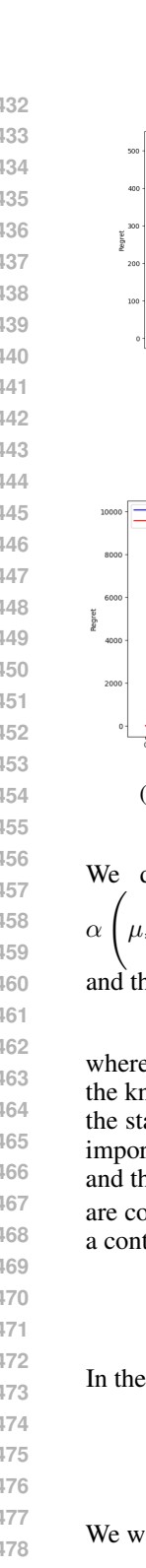

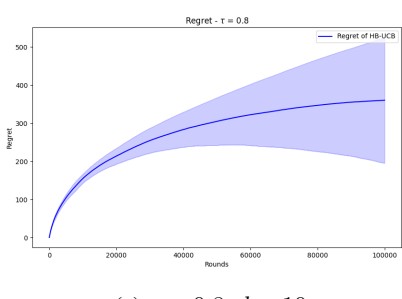

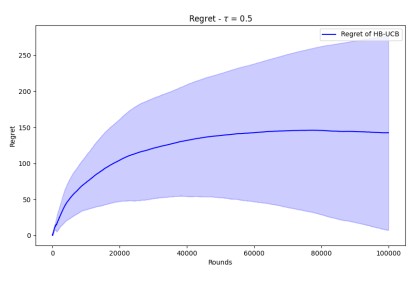

(a) $\tau = 0.8$, $d = 10$

(b) $\tau = 0.5$, $d = 10$

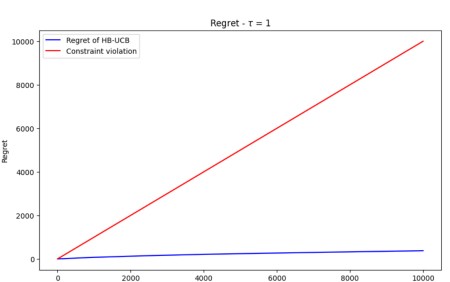

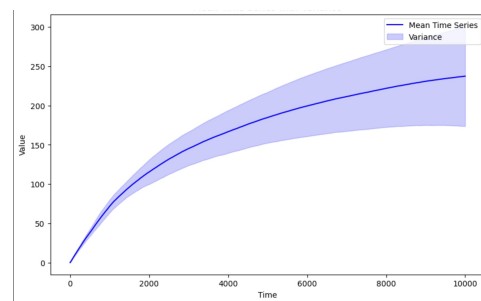

(c) Constraint violation in $10^4$ rounds.

(d) In cases where the model knows the noise distribution exactly.

We denote the set of feasible actions in round $t$ as $\mathcal{A}_t(X_t) = \{\alpha \in [0,1] \mid \alpha\left(\mu_*(X_t) + \gamma_c\sqrt{2\log(\frac{1}{\delta})}\right) \leq \tau\}$. The agent selects and action $\alpha_t \in \mathcal{A}_t(X_t)$. Now the reward and the cost signal take the following form.

$$R_t = \alpha_t\theta_*(X_t) + \alpha_t\xi_t^r, \quad C_t = \alpha_t\mu_*(X_t) + \alpha_t\xi_t^c$$

where $\theta_*(\cdot) \in \mathcal{G}_r$ and $\mu_*(\cdot) \in \mathcal{G}_c$ are the mean reward and cost function respectively that belong to the known function classes $\mathcal{G}_r, \mathcal{G}_c$. We will assume that $\theta_*(\cdot), \mu_*(\cdot)$ take values in $[-1,1]$, relaxing the standard assumption made that the non-linear functions take values in $[0,1]$. We show that the important property is that the non-linear functions remain bounded. We also assume that the reward and the cost signals are bounded, i.e. lie in $[-1,1]$. For the noise signals $\xi_t^r, \xi_t^c$ we assume that they are conditionally sub-Gaussian. Moreover, we use the definition of the width of a subset $\tilde{\mathcal{F}} \subset \mathcal{F}$ at a context $X \in \mathcal{A}$ by

$$w_{\tilde{\mathcal{F}}}(X) = \sup_{\underline{f}, \overline{f} \in \tilde{\mathcal{F}}} \left(\overline{f}(X) - \underline{f}(X)\right) \tag{12}$$

In the new terminology, the $T$ period regret is written as

$$\mathcal{R}(T, \pi) = \sum_{t=1}^{T} [\alpha_t^*\theta_*(X_t) - \alpha_t\theta_*(X_t)]$$

We want to apply the same regret decomposition as before. First, we define analogously $\hat{\mathcal{A}}_t(X_t) = \{\alpha \in [0,1] \mid \alpha\left(\mu_*(X_t) + \gamma_c\sqrt{2\log(\frac{1}{\delta})}\right) \leq \tau\}$. We also define

$$\alpha_t^* \in \arg\max_{\alpha \in \mathcal{A}_t} \theta_*(\alpha)$$

$$\alpha_t \in \arg\max_{\alpha \in \hat{\mathcal{A}}_t} \sup_{\theta \in \mathcal{G}_r} \theta(\alpha)$$

$$\tilde{\alpha}_t \in \arg\max_{\alpha \in \mathcal{A}_t} \sup_{\theta \in \mathcal{G}_r} \theta(\alpha)$$

As in **Proposition 1** in Russo & Van Roy (2013) our goal is to bound the regret using $w_{\tilde{\mathcal{F}}}(X_t)$. First we apply the same decomposition to express the regret in terms of the cost due to the lack of knowledge of $\theta_*$ and $\mu_*$.

$$\mathcal{R}(T, \pi) = \sum_{t=1}^{T} [\alpha^* \theta_*(X_t) - \tilde{\alpha}_t \theta_*(X_t)] + \sum_{t=1}^{T} [\tilde{\alpha}_t \theta_*(X_t) - \alpha_t \theta_*(X_t)]$$

The first sum can be bounded in a similar way to **proof A** in the appendix of Russo & Van Roy (2013). The second sum measures the regret the algorithm suffers from the lack of knowledge of $\mu$. Then we can bound in terms of $w_{\tilde{\mathcal{F}}_\mu}$ the same way as before.

**Lemma 7.1.** $\alpha_t^* \theta_*(X_t) - \tilde{\alpha}_t \theta_*(X_t) \leq w_{\mathcal{G}_r}(X_t) + 2\mathbb{1}(\theta_* \notin \mathcal{G}_r)$

*Proof.* The proof is in B.1. □

For the remaining part, we need to bound $|\tilde{\alpha}_t - \alpha_t|$ in terms of $\mu_*$. We will follow a similar proof as in the case of the inner product function.

**Lemma 7.2.** $|\tilde{\alpha}_t - \alpha_t| \leq w_{\mathcal{G}_c}(X_t)/\tau$

*Proof.* The proof is similar to the linear case and it is provided in B.2. □

Now that we have bound the regret in terms of the width of the set that the non-linear functions belong we can translate our results to bound for the regret. First, as in the linear model case, we define the reward and the cost set confidence radii as in (Pacchiano et al., 2024).

$$\rho_r(t, \delta') = 512 \log\left(\frac{24 \mid \mathcal{G}_r \mid \log(2t)}{\delta}\right), \quad \rho_c(t, \delta') = 512 \log\left(\frac{24 \mid \mathcal{G}_c \mid \log(2t)}{\delta}\right) \tag{13}$$

We also use the following notation $d_{eluder}^r = d_{eluder}(\mathcal{G}_r, 1/T)$ and $d_{eluder}^c = d_{eluder}(\mathcal{G}_c, 1/T)$. The algorithm is similar to that one in the linear case. Due to lack of space we write the algorithm in B.3. For the regret bound, like (Pacchiano et al., 2024), we use the Lemma 3 in (Chan et al., 2021), by setting $P = 1$.

**Theorem 7.1.** *With probability at least $1 - \delta'$, the regret of the algorithm B.3 satisfies*

$$\mathcal{R}(T) = \mathcal{O}(\sqrt{Td_{eluder}^r \rho_r(T, \delta'/2)} + 1/\tau \sqrt{Td_{eluder}^c \rho_c(T, \delta'/2)} + d_{eluder}^r + \frac{d_{eluder}^c}{\tau})$$

## 8 POSSIBLE EXTENSIONS

We hope that the community will recognize our conceptual contribution in modeling the optimal dose-finding problem. In the future, we anticipate that this model will be explored using real data from patients or cell cultures. Moreover, from a theoretical perspective, investigating instance-dependent bounds is an interesting direction, as is further studying the dependency of the regret on the confidence parameter $\delta$, which may itself be a function of $T$. Furthermore, since this algorithm processes medical data in its computations, it is important to consider the necessity of making it differentially private. Finally, it would be intriguing to solve the same problem using a Thompson sampling-like algorithm.

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

## A APPENDIX

### A.1 CONSTRAINT FORMULATION

We assume that the cost noise is conditionally sub-Gaussian with a known constant $\gamma_c$. Under this assumption, the random variable $\frac{C_t - \alpha_t \langle X_t, \mu \rangle}{\alpha_t}$ is $\gamma_c$ sub-Gaussian. We will define a constraint event such that, when satisfied, the cost signal remains below the threshold with high probability. Now, we can analyze the cost using the following theorem.

**Theorem A.1.** *[Sub Gaussian concentration bounds - Theorem 5.3 Lattimore & Szepesvári (2020)]*
*If $X$ is $\gamma_c$-subgaussian, then for any $\epsilon > 0$,*

$$\Pr(X \geq \epsilon) \leq \exp(-\frac{\epsilon^2}{2\gamma_c^2})$$

Using the above property of the cost noise and the theorem we derive that

$$\Pr(C_t \geq \tau \mid \mathcal{H}_t) = \Pr\left(\frac{C_t - \alpha_t\langle X_t, \mu^*\rangle}{\alpha_t} \geq \frac{\tau - \alpha_t\langle X_t, \mu^*\rangle}{\alpha_t} \,\middle|\, \mathcal{H}_t\right) \tag{14}$$

$$\leq \exp\left(-\frac{\left(\frac{\tau - \alpha_t\langle X_t, \mu^*\rangle}{\alpha_t}\right)^2}{2\gamma_c^2}\right) \tag{15}$$

By requiring the right-hand side to be less than or equal to $\delta$, we derive:

$$\exp\left(-\frac{\left(\frac{\tau - \alpha_t\langle X_t, \mu^*\rangle}{\alpha_t}\right)^2}{2\gamma_c^2}\right) \leq \delta$$

$$\frac{\left(\frac{\tau - \alpha_t\langle X_t, \mu^*\rangle}{\alpha_t}\right)^2}{2\gamma_c^2} \geq \log\left(\frac{1}{\delta}\right) \tag{16}$$

$$\frac{\tau - \alpha_t\langle X_t, \mu^*\rangle}{\alpha_t} \geq \gamma_c\sqrt{2\log\left(\frac{1}{\delta}\right)}$$

$$\tau \geq \alpha_t\left(\langle X_t, \mu^*\rangle + \gamma_c\sqrt{2\log\left(\frac{1}{\delta}\right)}\right)$$

## A.2 ANALYZING THE COST FOR APPROXIMATING $\theta$

The first term need to be bounded is $\sum_{t=1}^{T}(\alpha_t^* - \tilde{\alpha}_t)\langle X_t, \theta^*\rangle$. In order to bound this term we will follow a standard procedure in Linear Bandits. Initially, we will bound the term $\alpha_t^*\langle X_t, \theta^*\rangle$. With probability at least $1 - \delta$ it holds $\theta^* \in \mathcal{C}_t^\theta$, $\forall t \in [T]$.

$$\alpha_t^*\langle X_t, \theta^*\rangle \leq \max_{\theta \in \mathcal{C}_t^\theta}\{\alpha_t^*\langle X_t, \theta\rangle\}$$

$$\leq \max_{\alpha \in \mathcal{A}_t^f}\max_{\theta \in \mathcal{C}_t^\theta}\{\alpha\langle X_t, \theta\rangle\}$$

$$= \max_{\alpha \in \mathcal{A}_t^f}\{\alpha\langle X_t, \tilde{\theta}\rangle\}$$

$$= \tilde{\alpha}_t\langle X_t, \tilde{\theta}\rangle$$

Using the above it holds that

$$(\alpha_t^* - \tilde{\alpha}_t)\langle X_t, \theta^*\rangle \leq \tilde{\alpha}_t\langle X_t, \tilde{\theta} - \theta^*\rangle$$

## A.3 ANALYZING THE COST FOR APPROXIMATING $\mu^*$

We can bound the second term using the Cauchy-Schwarz inequality as follows:

$$(\tilde{\alpha}_t - \alpha_t)\langle X_t, \theta^*\rangle \leq |\tilde{\alpha}_t - \alpha_t| \cdot LS.$$

It remains to bound $|\tilde{\alpha}_t - \alpha_t|$. First, we remind the definitions of $\tilde{\alpha}_t$ and $\alpha_t$:

$$\tilde{\alpha}_t = \arg\max_{\alpha \in \mathcal{A}_t^f} \{\alpha \langle X_t, \hat{\theta}_t \rangle\},$$

$$\alpha_t = \arg\max_{\alpha \in \hat{\mathcal{A}}_t^f} \{\alpha \langle X_t, \hat{\theta}_t \rangle\}.$$

We observe that both the choice of $\tilde{\alpha}_t$ and the choice of $\alpha_t$ depend on the sign of the inner product $\langle X_t, \hat{\theta}_t \rangle$. If $\langle X_t, \hat{\theta}_t \rangle \geq 0$, then $\tilde{\alpha}_t$ equals the maximum element of the set $\mathcal{A}_t^f$. Similarly, $\alpha_t$ equals the maximum of the set $\hat{\mathcal{A}}_t^f$ when $\langle X_t, \hat{\theta}_t \rangle \geq 0$. On the other side, when $\langle X_t, \hat{\theta}_t \rangle < 0$, both $\tilde{\alpha}_t$ and $\alpha_t$ are zero.

We will write down again the sets $\mathcal{A}_t^f$ and $\hat{\mathcal{A}}_t^f$ to see the possible values for $(\tilde{\alpha}_t, \alpha_t)$:

$$\mathcal{A}_t^f = \left\{ \alpha \in [0,1] : \left( \langle X_t, \mu^* \rangle + \gamma_c \left( \sqrt{2 \log\left(\tfrac{1}{\delta}\right)} \right) \right) \alpha \leq \tau \right\},$$

$$\hat{\mathcal{A}}_t^f = \left\{ \alpha \in [0,1] : \left( \langle X_t, \tilde{\mu} \rangle + \gamma_c \left( \sqrt{2 \log\left(\tfrac{1}{\delta}\right)} \right) \right) \alpha \leq \tau \right\}.$$

Our estimator $\tilde{\mu}$ for $\mu^*$ is a pessimistic one. Among all possible choices for $\tilde{\mu}$, in order to be robust, we will choose $\tilde{\mu}$ such that $\hat{\mathcal{A}}_t^f$ has the smallest possible length.

Having that in mind, we have the four following scenarios for $(\tilde{\alpha}_t, \alpha_t)$:

1. $(1, 1)$

2. $\left( 1, \min\left( 1, \dfrac{\tau}{\gamma_c \left( \sqrt{2 \log\left(\tfrac{1}{\delta}\right)} \right) + \langle X_t, \tilde{\mu} \rangle} \right) \right)$

3. $\left( \min\left( 1, \dfrac{\tau}{\gamma_c \left( \sqrt{2 \log\left(\tfrac{1}{\delta}\right)} \right) + \langle X_t, \mu^* \rangle} \right), 1 \right)$

4. $\left( \min\left( 1, \dfrac{\tau}{\gamma_c \left( \sqrt{2 \log\left(\tfrac{1}{\delta}\right)} \right) + \langle X_t, \mu^* \rangle} \right), \min\left( 1, \dfrac{\tau}{\gamma_c \left( \sqrt{2 \log\left(\tfrac{1}{\delta}\right)} \right) + \langle X_t, \tilde{\mu} \rangle} \right) \right)$

For all the above cases we can show that

$$\max \mathcal{A}_t^f - \max \hat{\mathcal{A}}_t^f \leq \frac{\langle X_t, \tilde{\mu} - \mu^* \rangle}{\tau}.$$

Let's prove this one by one.

### A.3.1  1ST CASE

In this case, it is true that $\max \mathcal{A}_t^f - \max \hat{\mathcal{A}}_t^f = 1 - 1 = 0$.

### A.3.2  2ND CASE

The non-trivial pair in this case is $\left( 1, \dfrac{\tau}{\gamma_c \left( \sqrt{2 \log\left(\tfrac{1}{\delta}\right)} \right) + \langle X_t, \tilde{\mu} \rangle} \right).$

When the above relation for $\max \mathcal{A}_t^f$ and $\max \hat{\mathcal{A}}_t^f$ holds, then it is true that:

1. $\gamma_c \left( \sqrt{2 \log\left(\tfrac{1}{\delta}\right)} \right) + \langle X_t, \mu^* \rangle \leq 0,$

2. $\dfrac{\tau}{\gamma_c \left( \sqrt{2 \log \left( \frac{1}{\delta} \right)} \right) + \langle X_t, \tilde{\mu} \rangle} \leq 1.$

Using the above, we can bound $1 - \dfrac{\tau}{\gamma_c \left( \sqrt{2 \log \left( \frac{1}{\delta} \right)} \right) + \langle X_t, \tilde{\mu} \rangle}$ as follows:

$$
\begin{aligned}
1 - \frac{\tau}{\gamma_c \left( \sqrt{2 \log \left( \frac{1}{\delta} \right)} \right) + \langle X_t, \tilde{\mu} \rangle} &= \frac{\gamma_c \left( \sqrt{2 \log \left( \frac{1}{\delta} \right)} \right) - \tau + \langle X_t, \tilde{\mu} \rangle}{\gamma_c \left( \sqrt{2 \log \left( \frac{1}{\delta} \right)} \right) + \langle X_t, \tilde{\mu} \rangle} \\
&\leq \frac{-\langle X_t, \mu^* \rangle + \langle X_t, \tilde{\mu} \rangle}{\tau} \\
&= \frac{\langle X_t, \tilde{\mu} - \mu^* \rangle}{\tau}.
\end{aligned}
$$

### A.3.3  3RD CASE

We choose $\tilde{\mu}$ pessimistically, so in this case the only valid pair is $(1, 1)$ and $\mid \tilde{\alpha}_t - \alpha_t \mid = 0$.

### A.3.4  4TH CASE

This case can be divided into four different subcases:

1. $(1, 1)$ then $\mid \tilde{\alpha}_t - \alpha_t \mid = 0$.

2. $\left( 1, \dfrac{\tau}{\gamma_c \left( \sqrt{2 \log \left( \frac{1}{\delta} \right)} \right) + \langle X_t, \tilde{\mu} \rangle} \right)$. We saw that the above case can be bounded by $\dfrac{\langle X_t, \tilde{\mu} - \mu^* \rangle}{\tau}$.

3. $\left( \dfrac{\tau}{\gamma_c \left( \sqrt{2 \log \left( \frac{1}{\delta} \right)} \right) + \langle X_t, \mu^* \rangle}, 1 \right)$; this case cannot exist due to the way we choose $\tilde{\mu}$.

4. $\left( \dfrac{\tau}{\gamma_c \left( \sqrt{2 \log \left( \frac{1}{\delta} \right)} \right) + \langle X_t, \mu^* \rangle}, \dfrac{\tau}{\gamma_c \left( \sqrt{2 \log \left( \frac{1}{\delta} \right)} \right) + \langle X_t, \tilde{\mu} \rangle} \right)$.

In this case, it holds that $0 < \tau < \gamma_c \left( \sqrt{2 \log \left( \frac{1}{\delta} \right)} \right) + \langle X_t, \mu^* \rangle$ and $0 < \tau < \gamma_c \left( \sqrt{2 \log \left( \frac{1}{\delta} \right)} \right) + \langle X_t, \tilde{\mu} \rangle$.

We are going to bound $\mid \max \mathcal{A}_t^f - \max \hat{\mathcal{A}}_t^f \mid$ as follows:

$$
\begin{aligned}
\mid \max \mathcal{A}_t^f - \max \hat{\mathcal{A}}_t^f \mid &= \left| \frac{\tau}{\gamma_c \left( \sqrt{2 \log \left( \frac{1}{\delta} \right)} \right) + \langle X_t, \mu^* \rangle} - \frac{\tau}{\gamma_c \left( \sqrt{2 \log \left( \frac{1}{\delta} \right)} \right) + \langle X_t, \tilde{\mu} \rangle} \right| \\
&= \left| \frac{\tau \langle X_t, \tilde{\mu} - \mu^* \rangle)}{\left( \gamma_c \left( \sqrt{2 \log \left( \frac{1}{\delta} \right)} \right) + \langle X_t, \mu^* \rangle \right) \left( \gamma_c \left( \sqrt{2 \log \left( \frac{1}{\delta} \right)} \right) + \langle X_t, \tilde{\mu} \rangle \right)} \right| \\
&\leq \frac{\tau \langle X_t, \tilde{\mu} - \mu^* \rangle|}{\tau^2} \\
&= \frac{\langle X_t, \tilde{\mu} - \mu^* \rangle}{\tau}.
\end{aligned}
$$

### A.4 Proof of Theorem 5.1

*Proof.*

$$\mathcal{R}_{\mathcal{C}}(T) = \sum_{t=1}^{T} (\alpha_t^* - \tilde{\alpha}_t) \langle X_t, \theta^* \rangle + \sum_{t=1}^{T} (\tilde{\alpha}_t - \alpha_t) \langle X_t, \theta^* \rangle \tag{17}$$

$$\leq \sum_{t=1}^{T} |\tilde{\alpha}_t| \, \|x_t\|_{\Sigma_t^{-1}} \left\| \tilde{\theta} - \theta^* \right\|_{\Sigma_t} + \frac{LS}{\tau} \sum_{t=1}^{T} |\tilde{\alpha}_t| \, \|x_t\|_{\Sigma_t^{-1}} \, \|\tilde{\mu} - \mu^*\|_{\Sigma_t} \tag{18}$$

$$\leq \sum_{t=1}^{T} \beta_t(\delta', d) \, \|x_t\|_{\Sigma_t^{-1}} + \frac{LS}{\tau} \sum_{t=1}^{T} \beta_t(\delta', d) \, \|x_t\|_{\Sigma_t^{-1}} \tag{19}$$

$$\leq \beta_T(\delta', d)(1 + \frac{LS}{\tau}) \left( \sum_{t=1}^{T} \|x_t\|_{\Sigma_t^{-1}} \right) \tag{20}$$

$$\leq \beta_T(\delta', d)(1 + \frac{LS}{\tau}) \sqrt{2Td \log \left( 1 + \frac{TL^2}{\lambda} \right)} \tag{21}$$

$\square$

## B Non-Linear case

### B.1 Bound of $\alpha_t^* \theta_*(X_t) - \tilde{\alpha}_t \theta_*(X_t)$

*Proof.* The proof of 7.1 is similar to Russo & Van Roy (2013) as the decision set is the same for both $\alpha_t^*$ and $\tilde{\alpha}_t$. We define $U_t(\alpha) = \sup\{\alpha\theta_*(X_t) : \theta_* \in \mathcal{G}_r\}$ and $L_t(\alpha) = \inf\{\alpha\theta_*(X_t) : \theta_* \in \mathcal{G}_r\}$. When $\theta_*$ lies in $\mathcal{G}_r$ it holds that $L_t(\alpha) \leq \theta_*(\alpha) \leq U_t(\alpha)$. Using this we derive

$$\begin{aligned} \alpha_t^* \theta_*(X_t) - \tilde{\alpha}_t \theta_*(X_t) &\leq (U_t(\alpha_t^*) - L_t(\tilde{\alpha}_t)) \mathbb{1}(\theta_* \in \mathcal{G}_r) + 2\mathbb{1}(\theta_* \notin \mathcal{G}_r) \\ &\leq (U_t(\alpha_t^*) - L_t(\tilde{\alpha}_t)) + 2\mathbb{1}(\theta_* \notin \mathcal{G}_r) \\ &\leq w_{\mathcal{G}_r}(X_t) + 2\mathbb{1}(\theta_* \notin \mathcal{G}_r) + \underbrace{[U_t(\alpha_t^*) - U_t(\tilde{\alpha}_t)]}_{\leq 0 \text{ due to selection rule}} \end{aligned} \tag{22}$$

$\square$

Where in the last line we also used the fact that $\tilde{\alpha} \in [0, 1]$.

### B.2 Analyzing the cost for approximating $\mu_*(X_t)$

We need to bound $|\, \tilde{\alpha}_t - \alpha_t \,|$. First, we remind the definitions of $\tilde{\alpha}_t$ and $\alpha_t$:

$$\tilde{\alpha}_t = \arg \max_{\alpha \in \mathcal{A}_t^f} \{\alpha\hat{\theta}_*(X_t)\},$$

$$\alpha_t = \arg \max_{\alpha \in \hat{\mathcal{A}}_t^f} \{\alpha\hat{\theta}_*(X_t)\}.$$

We observe that both the choice of $\tilde{\alpha}_t$ and the choice of $\alpha_t$ depend on the sign of the value of $\hat{\theta}_*(X_t)$. If $\hat{\theta}_*(X_t) \geq 0$, then $\tilde{\alpha}_t$ equals the maximum element of the set $\mathcal{A}_t^f$. Similarly, $\alpha_t$ equals the maximum of the set $\hat{\mathcal{A}}_t^f$ when $\hat{\theta}_*(X_t) \geq 0$. On the other side, when $\hat{\theta}_*(X_t) < 0$, both $\tilde{\alpha}_t$ and $\alpha_t$ are zero.

We will write down again the sets $\mathcal{A}_t^f$ and $\hat{\mathcal{A}}_t^f$ to see the possible values for $(\tilde{\alpha}_t, \alpha_t)$:

$$\mathcal{A}_t^f = \left\{ \alpha \in [0, 1] : \left( \mu_*(X_t) + \gamma_c \left( \sqrt{2 \log \left( \tfrac{1}{\delta} \right)} \right) \right) \alpha \leq \tau \right\},$$

$$\hat{\mathcal{A}}_t^f = \left\{ \alpha \in [0, 1] : \left( \tilde{\mu}(X_t) + \gamma_c \left( \sqrt{2 \log \left( \tfrac{1}{\delta} \right)} \right) \right) \alpha \leq \tau \right\}.$$

Our estimator $\tilde{\mu}$ for $\mu^*$ is a pessimistic one. Among all possible choices for $\tilde{\mu}$, in order to be robust, we will choose $\tilde{\mu}$ such that $\hat{\mathcal{A}}_t^f$ has the smallest possible length.

Having that in mind, we have the four following scenarios for $(\tilde{\alpha}_t, \alpha_t)$:

1. $(1, 1)$

2. $\left( 1, \min \left( 1, \dfrac{\tau}{\gamma_c \left( \sqrt{2\log\left(\frac{1}{\delta}\right)} \right) + \tilde{\mu}(X_t)} \right) \right)$

3. $\left( \min \left( 1, \dfrac{\tau}{\gamma_c \left( \sqrt{2\log\left(\frac{1}{\delta}\right)} \right) + \mu_*(X_t)} \right), 1 \right)$

4. $\left( \min \left( 1, \dfrac{\tau}{\gamma_c \left( \sqrt{2\log\left(\frac{1}{\delta}\right)} \right) + \mu_*(X_t)} \right), \min \left( 1, \dfrac{\tau}{\gamma_c \left( \sqrt{2\log\left(\frac{1}{\delta}\right)} \right) + \tilde{\mu}(X_t)} \right) \right)$

For all the above cases we can show that

$$\max \mathcal{A}_t^f - \max \hat{\mathcal{A}}_t^f \le \frac{\tilde{\mu}(X_t) - \mu_*(X_t)}{\tau}.$$

Let's prove this one by one.

### B.2.1  1ST CASE

In this case, it is true that $\max \mathcal{A}_t^f - \max \hat{\mathcal{A}}_t^f = 1 - 1 = 0$.

### B.2.2  2ND CASE

The non-trivial pair in this case is $\left( 1, \dfrac{\tau}{\gamma_c \left( \sqrt{2\log\left(\frac{1}{\delta}\right)} \right) + \tilde{\mu}(X_t)} \right)$.

When the above relation for $\max \mathcal{A}_t^f$ and $\max \hat{\mathcal{A}}_t^f$ holds, then it is true that:

1. $\gamma_c \left( \sqrt{2\log\left(\frac{1}{\delta}\right)} \right) + \mu_*(X_t) \le 0$,

2. $\dfrac{\tau}{\gamma_c \left( \sqrt{2\log\left(\frac{1}{\delta}\right)} \right) + \tilde{\mu}(X_t)} \le 1.$

Using the above, we can bound $1 - \dfrac{\tau}{\gamma_c \left( \sqrt{2\log\left(\frac{1}{\delta}\right)} \right) + \tilde{\mu}(X_t)}$ as follows:

$$
\begin{aligned}
1 - \frac{\tau}{\gamma_c \left( \sqrt{2\log\left(\frac{1}{\delta}\right)} \right) + \tilde{\mu}(X_t)} &= \frac{\gamma_c \left( \sqrt{2\log\left(\frac{1}{\delta}\right)} \right) - \tau + \tilde{\mu}(X_t)}{\gamma_c \left( \sqrt{2\log\left(\frac{1}{\delta}\right)} \right) + \tilde{\mu}(X_t)} \\
&\le \frac{-\mu_*(X_t) + \tilde{\mu}(X_t)}{\tau} \\
&= \frac{\tilde{\mu}(X_t) - \mu_*(X_t)}{\tau}.
\end{aligned}
$$

### B.2.3 3RD CASE

We choose $\tilde{\mu}$ pessimistically, so in this case the only valid pair is $(1, 1)$ and $\mid \tilde{\alpha}_t - \alpha_t \mid = 0$.

### B.2.4 4TH CASE

This case can be divided into four different subcases:

1. $(1, 1)$ then $\mid \tilde{\alpha}_t - \alpha_t \mid = 0$.

2. $\left( 1, \dfrac{\tau}{\gamma_c \left( \sqrt{2 \log \left( \frac{1}{\delta} \right)} \right) + \tilde{\mu}(X_t)} \right)$. We saw that the above case can be bounded by $\dfrac{\tilde{\mu}(X_t) - \mu_*(X_t)}{\tau}$.

3. $\left( \dfrac{\tau}{\gamma_c \left( \sqrt{2 \log \left( \frac{1}{\delta} \right)} \right) + \mu_*(X_t)}, 1 \right)$; this case cannot exist due to the way we choose $\tilde{\mu}$.

4. $\left( \dfrac{\tau}{\gamma_c \left( \sqrt{2 \log \left( \frac{1}{\delta} \right)} \right) + \mu_*(X_t)}, \dfrac{\tau}{\gamma_c \left( \sqrt{2 \log \left( \frac{1}{\delta} \right)} \right) + \tilde{\mu}(X_t)} \right)$.

In this case, it holds that $0 < \tau < \gamma_c \left( \sqrt{2 \log \left( \frac{1}{\delta} \right)} \right) + \mu_*(X_t)$ and $0 < \tau < \gamma_c \left( \sqrt{2 \log \left( \frac{1}{\delta} \right)} \right) + \tilde{\mu}(X_t)$.

We are going to bound $\mid \max \mathcal{A}_t^f - \max \hat{\mathcal{A}}_t^f \mid$ as follows:

$$
\mid \max \mathcal{A}_t^f - \max \hat{\mathcal{A}}_t^f \mid = \left| \frac{\tau}{\gamma_c \left( \sqrt{2 \log \left( \frac{1}{\delta} \right)} \right) + \mu_*(X_t)} - \frac{\tau}{\gamma_c \left( \sqrt{2 \log \left( \frac{1}{\delta} \right)} \right) + \tilde{\mu}(X_t)} \right|
$$

$$
= \left| \frac{\tau \left( \langle X_t, \tilde{\mu} \rangle - \mu_*(X_t) \right)}{\left( \gamma_c \left( \sqrt{2 \log \left( \frac{1}{\delta} \right)} \right) + \mu_*(X_t) \right) \left( \gamma_c \left( \sqrt{2 \log \left( \frac{1}{\delta} \right)} \right) + \tilde{\mu}(X_t) \right)} \right|
$$

$$
\leq \frac{\tau \mid \tilde{\mu}(X_t) - \mu_*(X_t) \mid}{\tau^2}
$$

$$
= \frac{\tilde{\mu}(X_t) - \mu_*(X_t)}{\tau}.
$$

Now by following exaclty the same procedure as in 7.1 we derive that $|\tilde{\alpha}_t - \alpha_t| \leq w_{\mathcal{G}_c}(A)/\tau$.

### B.3 ALGORITHM FOR THE NON-LINEAR CASE

Due to limited space we analyze our algorithm for the Non-Linear case in the appendix. The algorithm is identical with the Linear case.

First we define the dataset $\mathcal{D}_t = \{(X_s, R_s, C_s)\}_{s=1}^{t-1}$ for $s$ such that $A_s \neq 0$, that is the dataset of observed information up to the beginning of round $t$, and $\|f\|_{\mathcal{D}_t} = \sqrt{\sum_{x \in \mathcal{D}_t} f^2(x)}$ the norm induced by the dataset for any function $f : \mathcal{A}_t \to \mathbb{R}$.

In every round we define the confidence ellipsoids as follows

$$\mathcal{C}_t^r(\delta) = \{\theta \in \mathcal{G}_r : \left\|\theta - \hat{\theta}\right\|_{\mathcal{D}_t} \leq \rho_r(t, \delta/2)\}$$

$$\mathcal{C}_t^c(\delta) = \{\theta \in \mathcal{G}_c : \left\|\theta - \hat{\theta}\right\|_{\mathcal{D}_t} \leq \rho_c(t, \delta/2)\}$$

Using these confidence intervals we compute the actions of the algorithm as follows. To compute the feasible dosages, first we solve the following Non-Linear program.

$$
\begin{aligned}
\max_{\boldsymbol{\mu}} \quad & \mu(X_t) \\
\text{subject to} \quad & \|\mu(X_t) - \hat{\mu}(X_t)\|_{\mathcal{D}_t} \leqslant b_t^2, \\
& \mu(X_t) + \gamma_c \sqrt{2 \log\left(\frac{1}{\delta}\right)} \geqslant 0
\end{aligned}
\tag{23}
$$

Then if there is no feasible solution in the above optimization problem we select $\hat{A}_t = [0, 1]$ otherwise, let say $\mathcal{K}(\hat{\mu}_t)$ its solution, then $\hat{\mathcal{A}}_t^f = [0, \frac{\tau}{\mathcal{K}(\hat{\mu}_t) + \gamma_c \sqrt{2 \log\left(\frac{1}{\delta}\right)}}]$ as before.

Our estimate for $\theta$ is $\tilde{\theta}(X_t) = max_{\theta \in \mathcal{C}_t^r(\delta')} \theta(X_t)$.

---

**Algorithm 2** Non-Linear High Probability Constrained UCB

---

1: **Input:** Constraint threshold $\tau \geq 0$; Confidence parameter $\delta$; Sub-Gaussianity constant $\gamma_c$
2: $\alpha_0 \leftarrow \min\{1, \frac{\tau}{\gamma_c \sqrt{2 \log\left(\frac{1}{\delta}\right)} + \max_X \mu_*(X)}\}$
3: **for** $t = 1, 2, \cdots, T$ **do**
4:     Compute $\hat{\mu}, \hat{\theta}$ by using Least Squares Estimators
5:     Construct the $\hat{\mathcal{A}}_t^f, \tilde{\theta}(X_t)$
6:     Compute action $\alpha_t = \arg\max_{\alpha \in \hat{\mathcal{A}}_t^f} \alpha \tilde{\theta}(X_t)$
7:     Take action $\alpha_t$ and if $\alpha_t \neq 0$ store the reward and the cost signals $(R_t, C_t)$
8: **end for**

---

