# OpenReview forum: "High Probability Contextual Bandits for Optimal Dosage Selection"
_ICLR.cc/2025/Conference — Submitted to ICLR 2025_

### Official Review · Reviewer_FL19 · 2024-10-24

**Soundness:** 3
**Presentation:** 4
**Contribution:** 3
**Rating:** 8
**Confidence:** 4

**Summary:**

The paper proposes a novel approach for determining optimal drug dosages using a linear contextual bandit model with stage-wise constraints. The key contribution is an algorithm that controls the toxicity of the administered dosage with high probability per step, rather than just in expectation as done in prior literature, thus addressing safety concerns in clinical settings. This method maximizes drug efficacy while minimizing the risk of overdose by ensuring toxicity remains below a threshold. The paper establishes theoretical regret bounds and demonstrates the algorithm's effectiveness through synthetic experiments, highlighting its potential in adaptive dose-finding scenarios.

**Strengths:**

+ The paper pushes the boundary of dose-finding methodology designs by considering the per-step toxicity constraint in a bandit-based solution. This setting has practical relevance and the solution thus can move the MAB-type methodology to be useful in practice. This aspect itself is very important not only to the ML community but probably more importantly, to the clinical trial methodology design community where safety and efficiency have been two of the critical considerations.
+ The high probability guarantee of cost violation is a novel component of this work. Prior works have only considered the average cost constraint, which as the authors argued may not be useful in practice. The algorithmic and theoretical aspects of linear contextual bandits are also of interest to other problems.
+ The trick to get around the problem of not having an initial safe dosage is clever.
+ The work makes some effort to extend the design to non-linear functions, with some initial results.

**Weaknesses:**

- One aspect that might be interesting to consider is that the toxicity tolerance threshold is often varying across patients. A more practical model would be to set $\tau$ also as a function of $X_t$.
- The argument in Section 4 relies on using $L$ and $S$ to construct the initial safe interval. This depends on the prior knowledge of accurate $L$ and $S$, which is, in some sense, reflecting the initialization of the trial. So fundamentally this is not surprising.
- The algorithm design part in Sec. 4 can be improved by more clearly articulating the differences to LinUCB.

**Questions:**

- Back to the issue of not having an initial safe dosage... why can't we just use the minimum dosage as the initial safe dosage?
- What is the theoretical novelty in deriving Theorem 5.1? How is it different than prior proofs of LinUCB or cost-expectation-based solutions?
- The simulations did not have any comparison to prior solutions. Can you add such comparisons? In particular, you have criticized prior solutions as only caring about expected cost constraint -- then how bad are they when you count step-wise constraint violation? What is the tradeoff of regret and constraint violation for all methods?

---

> ### Author Response · Authors · 2024-11-18
> **Responses to initial questions**
>
> We are grateful for your feedback and for sharing your insightful questions and remarks. Below are our reflections.
>
> Weaknesses:
>
> 1) We agree that the concept of making the toxicity threshold a function of $X_t$ is intriguing. This adjustment could likely be addressed within the second part of our paper, which focuses on the case of non-linear functions.
>
> 2) An upper bound on $L$ and $S$ is sufficient to determine the initial safe dosage. In practice, the vectors $X_t$, $\theta$, and $\mu$ are normalized, such that these bounds can be equal to one.
>
> 3) We agree with that commend, we work on that for the updated version.
>
> Questions:
> 1) Unfortunately, the minimum dosage is equal to zero and in that case we do not receive any feedback.
>
> 2) One technical difference is that the way to use the structure of the constraint to bound the cost term in the regret, that is the Lemma 5.2.
> In general, the main techniques from the cost-reward literature apply to our setting with some changes.
>
> 3) Since there is no previous work regarding high-probability constraint satisfaction, we plotted in graph (c) the constraint violations of the proposed algorithm from Pacchiano et al. (Contextual Bandits with Stage-Wise Constraints). Since in their algorithm the learner, at every round, chooses a vector that is not provided as input as in our algorithm, we made the following modification and concluded that, in the first $T$ rounds, their algorithm violates the constraint a linear number of times. In the updated version, we can simulate more setups to study this trade-off.

---

> > ### Author Response · Authors · 2024-11-22
> > **Reminder**
> >
> > Dear reviewer, we kindly request your attention to review our responses and share your valuable feedback. Your insights and guidance on these points are highly meaningful to us and will greatly contribute to advancing the discussion.

---

> > > ### Comment · Reviewer_FL19 · 2024-11-24
> > >
> > > I thank the authors for the responses. Overall I maintain my positive evaluation of this paper and keep my score unchanged.

---

### Official Review · Reviewer_Tfjc · 2024-11-01

**Soundness:** 3
**Presentation:** 3
**Contribution:** 3
**Rating:** 5
**Confidence:** 4

**Summary:**

The paper studies the optimal dosage finding problem with two objectives, maximize the drug’s efficacy and ensuring the toxicity especially from a high probability perspective. The authors adopt a linear contextual bandit formulation with stage-wise constraints, and design an efficient algorithm based on the idea of UCB. They establish a regret bound for this high-probability constrained approach, ensuring sublinear regret over time, meaning the model becomes more accurate as it learns.

**Strengths:**

1. The two-objective problem in dosage finding formulated by the paper is meaningful and practical relevant.
2. The way that how the authors adopt the idea of UCB looks interesting. The technical results are solid.
3. The paper is well written and relatively easy to follow.

**Weaknesses:**

1. Formulation. I am not sure whether it is common to think that the efficacy and the toxicity are linear with the dosage. From my experience in clinical trials, it is not usually the case.
2. Technical contribution. I like the authors' way of adpoting UCB, but the techniques seem very standard, mainly based on the repetitive use of the inequality from Abbasi-Yadkori et al., (2011). Is there any technical contribution that the authors want to highlight?
3. A very minor comment: I do not think you need to have a new paragraph only to say like "Proof. The proof is in A.3." You can save a lot of space to present more interesting results.

**Questions:**

An additional question to the weakness above, what does "stage-wise constraint" mean in the abstract? I can not see any explanation on this in the main text.

---

> ### Author Response · Authors · 2024-11-18
> **Responses to initial questions**
>
> We appreciate your feedback, as well as your insightful comments and questions. Please find our thoughts below.
>
> 1) We agree that the linear model is a simplified one.
> That is why we extended our results to any non-linear function that we can use to model the efficacy and the toxicity mechanisms.
>
> 2) Our differences with Abbasi-Yadkori et al., (2011) is the decomposition of the regret to express it as two terms, one induced by the reward and one by the cost signal.
> Some technical difficulties are the following:
>
>     1) Deriving a result that holds with high probability instead of in expectation.
>     2) Deriving an algorithm that learns efficiently the feasible set.
>     3) Bounding $(\tilde{\alpha}_t - \alpha_t)\langle X_t, \theta^* \rangle$
>
> 3) Thank you for that comment, this is definitely a good idea.
> We are going to adopt that change in the updated version.
>
> 4) The term "stage-wise constraint" in the bandits literature is that in every round a reward and a cost signal, $R_t, C_t$ respectively, are generated and the constraints are regarding to some function of the cost signal.

---

> > ### Author Response · Authors · 2024-11-22
> > **Reminder**
> >
> > Dear reviewer, we kindly request your attention to review our responses and share your valuable feedback. Your insights and guidance on these points are highly meaningful to us and will greatly contribute to advancing the discussion.

---

> > > ### Author Response · Authors · 2024-11-25
> > > **Reminder'**
> > >
> > > We thank the reviewer once again for their valuable comments and suggestions. If we have adequately addressed the questions and concerns raised, we kindly hope the reviewer may consider revisiting the score.

---

> ### Comment · Reviewer_Tfjc · 2024-11-26
> **Thank you for the response**
>
> Thank you for taking the time to provide such a thoughtful and clear response. I truly appreciate your effort. While your explanation is helpful, it does not substantially change my understanding of the contributions, especially when comparing with Abbasi-Yadkori et al., (2011). I believe that Abbasi-Yadkori et al., (2011) also provided high-probability bound. The design and the analysis are heavily depend on the inequality provided by Abbasi-Yadkori et al., (2011). Thus, I think it’s appropriate to maintain the current score.

---

### Official Review · Reviewer_Ck7p · 2024-11-04

**Soundness:** 4
**Presentation:** 4
**Contribution:** 3
**Rating:** 6
**Confidence:** 3

**Summary:**

The paper considers the problem regret version of the optimal dose finding with constraints that need to be satisfied with high probability.

**Strengths:**

The paper is well-written and solves a relevant problem. The paper is well-written and organized and is easy to follow.

**Weaknesses:**

The paper claims that is it the first to consider high-probability constraints on costs. I do not agree, since https://arxiv.org/pdf/2401.08016 ("Contextual Bandits with Stage-wise Constraints") seems to consider the setting with constraints that need to be satisfied with high probability as well. It is true that they need the knowledge of safe action, but I feel that data is benign in this application where there is likely to be a historical data set.

2)The motivation of the regret formulation is not clear. What does high regret mean ? Suppose the safety constraints were easy to satisfy, can then the algorithm just give out the maximum dosage? to make regret negative (Assuming the rewards are positive)

3)The techniques overall are standard in the linear function case. Maybe in future versions, the authors can specifically describe the main challenges faced in the proofs as a separate section

**Questions:**

Please see above

---

> ### Author Response · Authors · 2024-11-18
> **Responses to initial questions**
>
> Thank you for providing your valuable feedback and thoughtful questions and comments. Here is our response.
>
> 1) As we mention in our submission, in the paper the reviewer mentions ("Contextual Bandits with Stage-wise Constraints") the authors come up with an algorithm that ensures that with high probability the conditional expectations of the cost signals at every step are bounded. This is a much weaker notion of control than requiring the realizations of the cost signals to be bounded in all rounds. Controlling the expectation, even if with high probability means allowing for some realizations of the cost signal to be very large, and therefore dangerous for some patients.  Thus our results provide stronger safety guarantees than theirs.
>
> 2) It is true that past data can often be used to estimate an initial safe dosage.
> However, there are instances where such data are unavailable.
> For example, consider the emergence of a new illness, such as COVID-19, or the appearance of a novel pathogen that has recently crossed the species barrier, as seen in zoonotic viral diseases.
>
> 3) In bandit theory, we focus on ensuring sub-linear regret behavior so that, asymptotically, the error between the optimal policy and the policy to which we converge approaches zero.
> Another reason for minimizing regret is its equivalence to maximizing total efficacy.
> In the absence of constraints, the maximum permitted dosage should be assigned only when $\langle X_t, \theta^* \rangle > 0$.
> This necessitates learning the unknown vector $\theta^*$.
>
> 4) Some technical difficulties are the following:
>
>     1) Deriving a result that holds with high probability instead of in expectation.
>     2) Deriving an algorithm that learns efficiently the feasible set.
>     3) Bounding $(\tilde{\alpha}_t - \alpha_t)\langle X_t, \theta^* \rangle$

---

> > ### Author Response · Authors · 2024-11-22
> > **Reminder**
> >
> > Dear reviewer, we kindly request your attention to review our responses and share your valuable feedback. Your insights and guidance on these points are highly meaningful to us and will greatly contribute to advancing the discussion.

---

### Official Review · Reviewer_esg2 · 2024-11-04

**Soundness:** 2
**Presentation:** 2
**Contribution:** 2
**Rating:** 5
**Confidence:** 4

**Summary:**

This work investigates linear contextual bandits, with a brief exploration of a nonlinear case at the end, in the context of optimal dosage selection under a constraint function. A UCB-type algorithm is proposed to minimize regret while ensuring that the constraint is satisfied with high probability. Theoretical results are presented, and the approach is validated through numerical experiments.

**Strengths:**

This work introduces a new area of research: contextual bandits applied to dosage optimization while accounting for a (toxicity) constraint. The reviewer believes that this framework has broad practical relevance, making it a valuable and worthwhile topic to explore further.

**Weaknesses:**

The main concern raised by the reviewer is related to the assumptions. Specifically, the assumptions for the efficacy generation $R_t=\alpha_t <X_t,\theta^*> + \alpha_t\xi_t^r$ and $C_t=\alpha_t <X_t,\mu^*> + \alpha_t\xi_t^c$ are not intuitive to the reviewer.

Additionally, the presentation of the paper needs improvement for better readability, and the theoretical results should be expanded and explained in greater detail.

There are some minor comments:
$\gamma_\alpha$ is not defined. This should be stated in Assumption 1.
L134-138: not clear
L159-170: Thought X and Y are introduced, they are not explained. Also, the meaning of p_k and q_k should be stated.
L171-176: For the reviewer, it is hard to find some connection between this paragraph and the previous one.
The definition of K(x) should be more clear.
The citation format needs correction, and the language throughout the paper should be made more formal.

**Questions:**

1. Could the authors please provide justifications of the suggested efficacy and toxicity function by providing real-word examples?
2. In the regret, is it guaranteed that alpha_t is less than alpha_t^\star?

---

> ### Author Response · Authors · 2024-11-18
> **Responses to initial questions**
>
> Thank you for your feedback and your insightful comments and questions. Here are our thoughts.
>
> Weaknesses:
>
> 1) We adopted a linear model to formulate the reward and cost generating procedures for the following reasons.
> Firstly, linear models are widely used in the bandits literature, as mentioned in https://arxiv.org/pdf/2401.08016 and https://arxiv.org/pdf/2010.00081.
> The inner product in statistics can be interpreted as a measure of the correlation between the context $X_t$ and the unknown vector $\theta^*$.
> For example, consider the following scenario.
> $X_t$ may represent the results of medical tests of a patient before assigning a dose, as in chemotherapy.
> However, the doctor may not be able to precisely determine the significance of each characteristic (e.g. blood pressure, concentrations of specific substances in the blood, or specific pixels on an X-ray image) with respect to the efficacy of the drug for the patient.
> In this context, $\theta^*$ represents a probability distribution vector that captures the initially unknown importance of each feature.
> In this case, if the inner product $\langle X_t, \theta^* \rangle > 0$, it indicates that the patient's medical condition is positively correlated with the disease, suggesting that the patient suffers from that disease.
> When the patient suffers from the disease, increasing the dosage (up to the allowed threshold) enhances the therapeutic effect.
> However, increasing the dose also amplifies both the toxicity and the noise, meaning that higher doses intensify the randomness of the drug's effect on the patient.
> Therefore, we need to balance these two aspects.
> In the second section, we extend our results to any function of the context $X_t$.
> In this case, such a function could be a neural network trained to learn the importance of medical records for disease diagnosis, or any nonlinear function established in the medical literature.
>
> 2) We really appreciate all reviewers feedback and we are working on incorporating it to our manuscript.
>
> 3) Regarding $\gamma_{\alpha}$, do you mean $\gamma_c$, or $\gamma_r$?
> The positive real-valued constants $\gamma_r$ and $\gamma_c$ characterize the measure of sub-Gaussianity of a random variable. This property describes the decay rate of the tails of the random variable's distribution, analogous to how the standard deviation characterizes the spread in a Gaussian distribution.
> For a formal definition of sub-Gaussian random variables, we refer readers to classical texts in bandit theory and high-dimensional statistics such as:
>
> i) Lattimore, T.,  Szepesvári, C. (2020). \textit{Bandit Algorithms}. Cambridge University Press.
>
> ii) Vershynin, R. (2018). \textit{High-Dimensional Probability: An Introduction with Applications in Data Science}. Cambridge University Press.
>
> About $\mathcal{K}(x)$, we can equivalently write it as $\mathcal{K}(x) =$ { $\mu \in \mathcal{C}^t_\mu \mid \mu \\ \text{ belongs to the feasible solutions of (6) with input }x$ }.
>
> Questions:
>
> 1) The assignment of any drug to a patient induces both positive and negative effects.
> The positive effect reflects the drug's efficacy, while the negative effect pertains to its toxicity.
> We employed a contextual approach, recognizing that the effects of a drug depend on the patient's phenotype—specific characteristics measurable through medical tests.
> A linear model was chosen to evaluate the correlation between the patient’s phenotype and the significance of each feature in the mechanisms generating reward (efficacy) and toxicity.
> Linear models are the simplest in statistical analysis, and simpler models tend to generalize better.
> However, when additional structural information about the mechanisms of efficacy and toxicity becomes available, the approach can be extended to non-linear functions.
> Any established non-linear function in the literature can then be employed to better capture the relationship between the patient's condition and the mechanisms determining the reward-to-cost ratio.
>
> 2) In bandit theory the regret can take negative values too, so it does not matter if $\alpha_t \leq \alpha_t^*$.
> We only care about the sub-linear behavior of the regret.
> Without the cost signal constraint, it is guaranteed with high probability over the past realizations of the noise that $\alpha_t \geq \alpha_t^*$ as we choose $\alpha_t$ optimistically over the possible values of $\theta^*$.
> Although, taking into account the cost-toxicity constraint, we are initially robust in our choices for $\alpha_t$ so, during the first rounds it happens that $\alpha_t \leq \alpha_t^*$.

---

> ### Author Response · Authors · 2024-11-22
> **Reminder**
>
> Dear reviewer, we kindly request your attention to review our responses and share your valuable feedback. Your insights and guidance on these points are highly meaningful to us and will greatly contribute to advancing the discussion.

---

> > ### Comment · Reviewer_esg2 · 2024-11-22
> > **Linear Assumption for efficacy and toxicity**
> >
> > Thank you to the authors for their detailed responses.
> >
> > The reviewer agrees that the linear assumptions for reward and toxicity are reasonable in many cases but believes they require more rigorous justification. For instance, defining $X_t, \theta$ as the efficacy of a medicine per unit implies that efficacy increases linearly with dosage. While this assumption seems plausible, it needs further elaboration and support.
> >
> > Another issue is the assumption that the variance of the error term increases quadratically with dosage. The reviewer finds this assumption less convincing than the linear effect assumption, even if an increasing variance over dosage is reasonable. A constant variance of error might be a more appropriate choice in many cases. While a quadratic growth rate might appear overly extreme, the reviewer believes the quality of the work would improve significantly if both of these assumptions were more thoroughly justified with real-world examples.
> >
> > The reviewer suggests that the authors dedicate a substantial portion of the work to justifying these assumptions. Without stronger justification, readers may find it difficult to fully grasp and accept these novel linearity assumptions. As some concerns have been addressed, score adjustment will be made.

---

### Author Response · Authors · 2024-11-30

We sincerely thank all the reviewers for their valuable comments, constructive discussions, and dedicated efforts. Their input has been instrumental in significantly improving our work.

---

### Meta-Review · Area_Chair_AQrN · 2024-12-18

**Metareview:**

This paper studies a bandit problem where the agent chooses drug dosage to maximize reward, which is a linear function of the dosage; and controls cost with a high probability, which is also a linear function of the dosage. Both the reward and cost are linear functions of feature vectors and unknown model parameters. The authors propose an optimistic algorithm, analyze it, and evaluate it in simulations. The scores of this paper are 8, 6, and 2x 5, which is an improvement over the initial 8, 6, 5, and 3. The reviewers had multiple concerns:

* **Modeling assumptions:** Some modeling assumptions are highly stylized. For instance, while the linearity of reward and toxicity is reasonable in many cases, this needs to be rigorously justified. Defining the efficacy of a medicine per unit implies that the efficacy increases linearly with dosage. The assumption that the variance of the error term increases quadratically with dosage is unrealistic.

* **Technical challenge:** The reviewers are unconvinced that this paper is technically challenging. The high-probability confidence intervals used in this work have been known for a decade.

* **Experiments:** A strong experimental section, for instance using off-policy evaluation on real patient data, would carry this paper. Unfortunately, all experiments in this paper are synthetic.

These concerns require a major revision and therefore the paper cannot be accepted at this time.

**Additional Comments On Reviewer Discussion:**

See the meta-review for details.

---

### Decision · Program_Chairs · 2025-01-22

Reject